# AGGREGATION OF MULTI DIFFUSION MODELS FOR ENHANCING LEARNED REPRESENTATIONS

## ABSTRACT

Diffusion models have achieved remarkable success in image generation, particularly with the various applications of classifier-free guidance conditional diffusion models. While many diffusion models perform well when controlling for particular aspect among style, character, and interaction, they struggle with fine-grained control due to dataset limitations and intricate model architecture design. This paper introduces a novel algorithm, Aggregation of Multi Diffusion Models (AMDM), which synthesizes features from multiple diffusion models into a specified model, enhancing its learned representations to activate specific features for fine-grained control. AMDM consists of two key components: spherical aggregation and manifold optimization. Spherical aggregation merges intermediate variables from different diffusion models with minimal manifold deviation, while manifold optimization refines these variables to align with the intermediate data manifold, enhancing sampling quality. Experimental results demonstrate that AMDM significantly improves fine-grained control without additional training or inference time, proving its effectiveness. Additionally, it reveals that diffusion models initially focus on features such as position, attributes, and style, with later stages improving generation quality and consistency. AMDM offers a new perspective for tackling the challenges of fine-grained conditional control generation in diffusion models: We can fully utilize existing conditional diffusion models that control specific aspects, or develop new ones, and then aggregate them using the AMDM algorithm. This eliminates the need for constructing complex datasets, designing intricate model architectures, and incurring high training costs. Code is available at: `https://github.com/Hammour-steak/AMDM`.

## 1 INTRODUCTION

Diffusion models (Sohl-Dickstein et al., 2015; Ho et al., 2020; Song et al., 2021a;b; Karras et al., 2022) are designed to establish a relationship between data and noise, utilizing neural networks to learn the reverse process (Anderson, 1982). This enables the generation of data from random noise, showcasing exceptional performance in generative tasks. In practical applications like Text-to-Image (T2I) (Nichol et al., 2022; Chen et al., 2023; Lee et al., 2024; Xu et al., 2024) and Image-to-Image (I2I) (Zhang et al., 2023; Mou et al., 2024) generation, conditional diffusion models (Rombach et al., 2022; Chung et al., 2023; Esser et al., 2024) are widely used. These models achieve state-of-the-art results and provide highly flexible conditional control, making them a central focus in current research.

Recent research on conditional diffusion models has focused on achieving fine-grained control over image generation. However, maintaining consistency across diverse nuanced control, including object attributes (Wu et al., 2023a; Wang et al., 2024a), interactions (Hoe et al., 2024; Jia et al., 2024), layouts (Zheng et al., 2023; Chai et al., 2023; Chen et al., 2024b), and style (Wang et al., 2023; Huang et al., 2024; Qi et al., 2024), remains a significant challenge. Generating multiple objects with overlapping bounding boxes can lead to attribute leakage, where one object's description inappropriately influences others, causing inconsistencies between objects and the background. Fine-grained interaction details may be illogical, and style integration may compromise object attributes.

Existing approaches only partially address these issues due to the inherent complexity and diversity of fine-grained control, coupled with limitations in datasets and model architec-

tures. Some works (Li et al., 2023; Zhou et al., 2024; Wang et al., 2024b) may perform well in preventing attribute leakage among multiple instances during layout generation but perform poorly in managing object interactions, while others (Ye et al., 2023; Huang et al., 2024) may excel in style transfer but exhibit limited control over layout.

Interestingly, most of these methods are based on Stable Diffusion (Rombach et al., 2022), which is theoretically grounded in the DDPM (Ho et al., 2020) and classifier-free guidance (Dhariwal & Nichol, 2021) for conditional control. Therefore, for these conditional diffusion models grounded in the same theoretical foundation, our objective is to overcome this challenge by developing a method that effectively aggregates the advantages of each model, leveraging their unique strengths to achieve fine-grained control.

This paper proposes a novel algorithm called Aggregation of Multi Diffusion Models (AMDM), as shown in Figure 1. AMDM aggregates intermediate variables from different conditional diffusion models which based on the same theoretical foundation, into a specific model during inference. This approach enhances learned representations by absorbing characteristics from different models, regardless of variations in architecture or training datasets, without requiring additional training, avoiding the need for complex datasets and intricate model designs. Our experiments show that our proposed algorithm AMDM

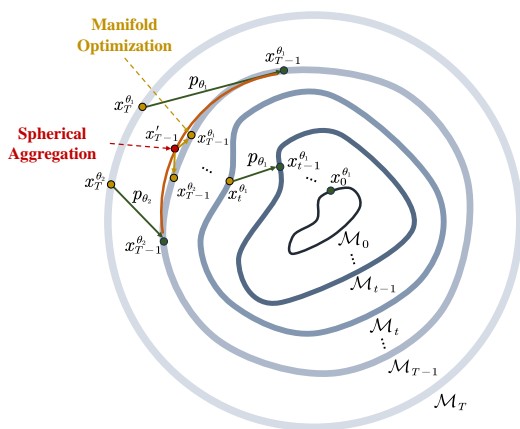

Figure 1: Geometry of AMDM. The process of aggregating features from model $p_{\theta_2}$ into model $p_{\theta_1}$ involves two main stages: spherical aggregation and manifold optimization. The AMDM algorithm is utilized to incorporate conditional information during the initial steps of the sampling process. Direct sampling is then applied to expedite the process and generate high-quality images.

significantly improves the fine-grained generation capability of a specific conditional diffusion model. Furthermore, it demonstrate that diffusion models with a shared theoretical foundation possess the same mathematical essence, allowing operations on their intermediate variables, while also revealing a phenomenon where early sampling steps focus on generating diverse features, and later stages prioritize quality and consistency.

Our main contributions are as follows:

- We propose a novel diffusion model aggregation algorithm AMDM that can aggregate intermediate variables from different conditional diffusion models of the same theoretical foundation, absorbing the characteristics of each model and enabling the generation of fine-grained control tasks.

- We conduct a variety of experiments by aggregating different conditional diffusion models, and both visual and quantitative results demonstrate noticeable improvements in areas where the models previously exhibited weaker control, validating the effectiveness of the algorithm.

- Our algorithm and experiments reveal some unique properties of diffusion models: Diffusion models with a shared theoretical foundation possess the same mathematical essence, even if they differ in architecture, allowing operations on their intermediate variables; Furthermore, diffusion models initially focus on the generation of features such as position, attributes, and style, while later stages emphasize generation quality and consistency.

## 2 PRELIMINARIES

### 2.1 DIFFUSION MODEL

Diffusion models are a class of generative models that progressively add noise to guide the data distribution $q(\mathbf{x}_0)$ towards a Gaussian distribution. By employing maximum likelihood estimation

through neural networks, diffusion models learn a reverse process, enabling them to generate data by progressively denoising from arbitrary noise. In the classical DDPM (Ho et al., 2020) formulation, the noise addition process from time $t - 1$ to $t$ is defined as:

$$q(\mathbf{x}_t|\mathbf{x}_{t-1}) = \mathcal{N}(\sqrt{\alpha_t}\mathbf{x}_{t-1}, (1 - \alpha_t)\boldsymbol{I}), \tag{1}$$

where, $\mathbf{x}_t$ represents the noisy data at timestep $t \in [0, T]$, $\alpha_t$ is the coefficient drift schedule generally satisfying $\lim_{t \to T} \alpha_t = 0$. From equation 1, we can readily derive the forward marginal distribution:

$$q(\mathbf{x}_t|\mathbf{x}_0) = \mathcal{N}(\sqrt{\bar{\alpha}_t}\mathbf{x}_0, (1 - \bar{\alpha}_t)\boldsymbol{I}), \tag{2}$$

where $\bar{\alpha}_t = \prod_{i=1}^t \alpha_i$. Equation 2 indicates that we can directly obtain $\mathbf{x}_t$ from $\mathbf{x}_0$ avoiding multistep sampling. Assuming the generative model is $p_\theta(x_0)$, consider the variational lower bound of its likelihood as the loss function, i.e., the KL divergence of the joint probability:

$$\begin{aligned} \mathcal{L} &= KL(q(\mathbf{x}_{0:T})\|p_\theta(\mathbf{x}_{0:T})) \\ &\propto \mathbb{E}_{t,\mathbf{x}_t,\epsilon_t}\left[\|\boldsymbol{\epsilon}_t - \boldsymbol{\epsilon}_\theta(\mathbf{x}_t, t)\|^2\right], \end{aligned} \tag{3}$$

where $\boldsymbol{\epsilon}_t \sim \mathcal{N}(\mathbf{0}, \boldsymbol{I})$ and $\boldsymbol{\epsilon}_\theta(\mathbf{x}_t, t)$ is the denoising neural network. More generally, we utilize DDIM (Song et al., 2021a) sampling which directly defines the forward process equation 2 compared to DDPM. Ultimately, the reverse sampling process as follows:

$$p_\theta(\mathbf{x}_{t-1} \mid \mathbf{x}_t) = \mathcal{N}\left(\sqrt{\frac{\bar{\alpha}_{t-1}}{\bar{\alpha}_t}}\mathbf{x}_t + \left(\sqrt{1 - \bar{\alpha}_{t-1} - \sigma_t^2} - \sqrt{\frac{\bar{\alpha}_{t-1}(1 - \bar{\alpha}_t)}{\bar{\alpha}_t}}\right)\boldsymbol{\epsilon}_\theta(\mathbf{x}_t, t), \sigma_t^2\boldsymbol{I}\right), \tag{4}$$

where $\sigma_t$ is a free variable. When $\sigma_t^2 = \frac{1 - \bar{\alpha}_{t-1}}{1 - \bar{\alpha}_t}(1 - \frac{\bar{\alpha}_t}{\bar{\alpha}_{t-1}})$, it corresponds to DDPM sampling, while $\sigma_t^2 = 0$ results in a deterministic sampling, which is the origin of the name DDIM.

## 2.2 CLASSIFIER-FREE GUIDANCE

The key of conditional control in diffusion models is estimating $q(\mathbf{x}_{t-1}|\mathbf{x}_t, y)$ given the condition $y$. In unconditional generation, according to equation 4, it can be written as:

$$p_\theta(\mathbf{x}_{t-1} \mid \mathbf{x}_t) = \mathcal{N}(\boldsymbol{\mu}_\theta(\mathbf{x}_t, t), \sigma_t^2\boldsymbol{I}). \tag{5}$$

Accordingly, Classifier-Free Guidance (Dhariwal & Nichol, 2021) directly incorporates the condition $y$ into the mean for estimation:

$$\begin{aligned} p_\theta(\mathbf{x}_{t-1} \mid \mathbf{x}_t, y) &= \mathcal{N}(\boldsymbol{\mu}_\theta(\mathbf{x}_t, t, y), \sigma_t^2\boldsymbol{I}) \\ &= \mathcal{N}\left(\sqrt{\frac{\bar{\alpha}_{t-1}}{\bar{\alpha}_t}}\mathbf{x}_t + \left(\sqrt{1 - \bar{\alpha}_{t-1} - \sigma_t^2} - \sqrt{\frac{\bar{\alpha}_{t-1}(1 - \bar{\alpha}_t)}{\bar{\alpha}_t}}\right)\boldsymbol{\epsilon}_\theta(\mathbf{x}_t, t, y), \sigma_t^2\boldsymbol{I}\right). \end{aligned} \tag{6}$$

Therefore, the training loss function is:

$$\mathcal{L} \propto \mathbb{E}_{t,\mathbf{x}_t,\epsilon_t}\left[\|\boldsymbol{\epsilon}_t - \boldsymbol{\epsilon}_\theta(\mathbf{x}_t, t, y)\|^2\right]. \tag{7}$$

The noise model $\boldsymbol{\epsilon}_\theta(\mathbf{x}_t, t, y)$ incorporates conditioning on $y$, guiding the denoising process towards the conditioned direction, thereby enabling conditional sampling and generation.

## 3 AMDM

In this section, we first analyzes the challenges and limitations of current fine-grained control research, providing a general direction for potential solutions. Next, a rigorous formal definition of the research problem is presented for clarity. Finally, we describe the design principles and propose the AMDM algorithm.

### 3.1 ANALYSIS

Current fine-grained conditional control models tend to have limited control capabilities and face numerous issues. For example, given the caption "A red hair girl is drinking from a blue bottle of water, oil painting" and corresponding bounding boxes for positioning control, different models are likely to show varying performance, as illustrated in Figure 2. Model A, which receives additional inputs for position information and actions, excels in generating high-quality generation of positioning and interactions. However, it struggles with attribute control and maintaining the oil painting style. Conversely, Model B incorporates extra input for position and attribute information, managing both but not accurately capture interactions and stylistic elements.

Model C references the style of an image, enabling precise management of style characteristics but lacking adequate control over location and attribute details. The fundamental reason for these issues lies in the complexity and flexibility of fine-grained control tasks, which makes it challenging for limited datasets and specific model architectures to account for all the intricate features. While implementing a specific feature for a particular task is relatively straightforward, integrating these features for fine-grained control remains a significant challenge.

*Caption:* A **red** hair girl is drinking a **blue** bottle of water, **oil painting**

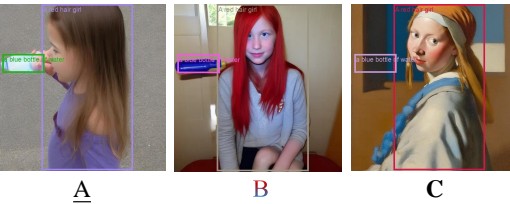

Figure 2: Examples of fine-grained conditional control of the same caption by different models.

While different models may have varying additional input conditions, these conditions are often mutually compatible, leading to consistent objectives, which allows the models to generate similar images within their respective generation domains. Theoretically, model A has the capability to generate images that meet all the composite conditions of the caption for fine-grained control, although a single sampling might not fully activate this capability. It is noteworthy that these models share a common theoretical foundation, as they are all predicated on the same diffusion process and Classifier-Free Guidance (CFG) conditional control mechanism. Recognizing this shared basis, our objective is to develop an algorithm that leverages these commonalities to integrate the distinctive characteristics of multiple models into a specific model, achieving fine-grained conditional control in a more direct and efficient manner.

### 3.2 DEFINITION

It is necessary to formally define the above concepts to facilitate a more rigorous exposition of the research problem.

**Definition 1.** *For the diffusion model $p_\theta$, its **generation domain** of $t$ under condition $y$ is:*

$$D_{t,y}^\theta = \{\mathbf{x}_t \in \mathbb{R}^n \mid \mathbf{x}_t \sim p_\theta(\mathbf{x}_t|y)\}. \tag{8}$$

When $t = 0$, $D_{0,y}^\theta$ represents the set of all possible data generated by the diffusion model $p_\theta$ under the condition $y$. Assuming the $n$-dimensional data generated by $p_\theta$ resides on an $m$-dimensional manifold $\mathcal{M}_0 = \{x_0 \in \mathbb{R}^n, x_0 \sim p_\theta(x_0)\}, m \ll n$, the data in $D_{0,y}^\theta$ will reside on a lower $l$ dimensional ($l < n$) submanifold due to the constraints imposed by $y$. When $t \neq 0$, Chung et al. (2022) have proved that $\mathcal{M}_t = \{\mathbf{x}_t \in \mathbb{R}^n, \mathbf{x}_t \sim p_\theta(\mathbf{x}_t)\}$ is an $(n-1)$-dimensional intermediate data manifold, which approximates an $n$-dimensional hypersphere when $t$ is large. Additionally, the introduction of the condition $y$ does not affect the forward noise addition process. As a result, we can infer through a similar proof that the data in $D_{t,y}^\theta$ will also reside on an $(n-1)$-dimensional manifold.

**Definition 2.** *For a set of different diffusion models $M = \{p_{\theta_1}, p_{\theta_2}, \ldots, p_{\theta_N}\}$, and corresponding conditions $Y = \{y_1, y_2, \ldots, y_N\}$, if $\bigcap_{i=1}^n D_{t,y_i}^{\theta_i} \neq \emptyset$, then $M$ is referred to as the **t-compatible model set** under condition $Y$.*

This indicates that there exists an intersection on the manifold where $x_t$ resides under different conditions corresponding to various diffusion models within $M$. Evidently, the points at these intersections encapsulate all the information of the condition $Y$. Therefore, intermediate variables $x_t$ of any model encapsulate all conditional information, enabling the generation of images under additional composite conditions.

For a specific fine-grained control task, although different models may accept varying additional input types, they share a common objective. If this task is decomposed into different conditions $y_1, y_2, \ldots, y_n$ recognizable by different models, there must exist intermediate variables $x_t$ that satisfy each of these conditions for any $t \in [0, T]$. Consequently, these models collectively form a $t$-compatible model set under the condition $Y = \{y_1, y_2, \ldots, y_n\}$. Given this, we have the following assumption:

**Assumption 1.** *If $y_1, y_2, \ldots, y_n$ all describe a specific task, $M = \{p_{\theta_1}, p_{\theta_2}, \ldots, p_{\theta_N}\}$ forms a $t$-compatible model set under the condition $Y = \{y_1, y_2, \ldots, y_n\}$ for any $t \in [0, T]$.*

In practical applications, we focus on generation for a specific task. Based on this assumption, these models form a $t$-compatible model set under the task condition $Y$, enabling generation under composite conditions and achieving fine-grained control.

### 3.3 ALGORITHM

For simplicity, we begin by considering the case of two models, specifically focusing on aggregating the conditional control information from $p_{\theta_2}$ into $p_{\theta_1}$.

**Spherical Aggregation.** Sampling $x_0$ from the model $p_{\theta_1}$ involves a series of reverse diffusion steps: $p(T) \sim N(\mathbf{0}, \mathbf{I})$, $p_{\theta_1}(\mathbf{x}_{T-1}^{\theta_1}|\mathbf{x}_T^{\theta_1}, y)$, ..., $p_{\theta}(x_0^{\theta_1}|x_1^{\theta_1}, y_1)$, signifying that $x_t^{\theta_1} \in D_{t,y_1}^{\theta_1}$ for all $t \in [0, T]$. It indicates that the new intermediate variable must also reside within $D_{t,y_i}^{\theta_1}$ for the sampling of $p_{\theta_1}$, after aggregating another model $p_{\theta_2}$. Regarding the aggregation of conditional information, we propose two key design points for the algorithm: 1) a shared latent space encoder and same diffusion process; and 2) spherical linear interpolation for conditional control information aggregation.

For the first point, the alignment of the latent space ensures the consistency of the initial data manifold $\mathcal{M}_0$, while maintaining an identical diffusion process preserves the consistency of intermediate data manifolds $\mathcal{M}_t$ ($t > 0$). This guarantees that all operations performed on different intermediate variables remain closed within the corresponding manifolds $\mathcal{M}_t$ for any $t$. For the second point, since the noisy $n$-dimensional data $\mathbf{x}_t$ resides on the manifold of an approximate $(n-1)$-dimensional hypersphere (Chung et al., 2022), using spherical linear interpolation for aggregation maximizes the retention of the aggregated data on the original manifold, minimizing deviations:

$$\mathbf{x}'_{t-1} = w\mathbf{x}_{t-1}^{\theta_1} + \sqrt{1-w^2}\mathbf{x}_{t-1}^{\theta_2}, \tag{9}$$

where $\mathbf{x}'_{t-1}$ represents the aggregated intermediate variable, $\mathbf{x}_{t-1}^{\theta_1}$ and $\mathbf{x}_{t-1}^{\theta_2}$ are sampled from $p_{\theta_1}(\mathbf{x}_{t-1}^{\theta_1}|\mathbf{x}_t^{\theta_1}, y_1)$ and $p_{\theta_2}(\mathbf{x}_{t-1}^{\theta_2}|\mathbf{x}_t^{\theta_2}, y_2)$ respectively, and $w \in [0, 1]$ is the weighting factor that balances the contribution of each model. Spherical aggregation integrates the conditional control information of $p_{\theta_2}$ into $p_{\theta_1}$, while remaining the new variables stable near the manifold.

**Manifold Optimization.** Ideally, $x'_{t-1}$ would reside on $D_{t-1,y_1}^{\theta_1} \bigcap D_{t-1,y_2}^{\theta_2}$, achieving the aggregation of conditional information from different models. However, deviations are likely to occur in practice, leading us to propose a manifold optimization algorithm to correct the deviation of the aggregated data on the manifold. Considering the step from $t$ to $t-1$, since $p_{\theta_1}(\mathbf{x}_{t-1}^{\theta_1}|\mathbf{x}_t^{\theta_1}, y_1)$ follows a normal distribution, a single sample is likely to be drawn near a peak with high confidence. This suggests that the true data consistently resides near the peak and on $D_{t,y_1}^{\theta_1}$. Consequently, we can shift $x'_{t-1}$ along the gradient of the probability density function $p_{\theta_1}(\mathbf{x}_{t-1}^{\theta_1}|\mathbf{x}_t^{\theta_1}, y_1)$ by performing gradient ascent, adjusting the value to position

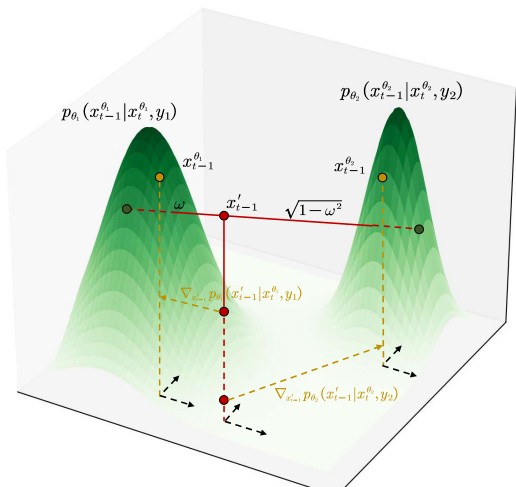

Figure 3: Geometry of manifold optimization between two models. The green points represent the original sampled points, the red points indicate the results of spherical aggregation, and the gold points denote the final results after manifold optimization.

it near the peak and bringing it back to $D_{t-1,y_1}^{\theta_1}$. In light of this, we propose the main proposition of the manifold optimization algorithm:

**Proposition 1.** *For the diffusion model $p_{\theta_1}$ and any new intermediate variable $\mathbf{x}'_{t-1}$, let:*

$$\mathbf{x}_{t-1}^{\theta_1} = \mathbf{x}'_{t-1} - \eta_{t-1}^{\theta_1} \frac{\mathbf{x}'_{t-1} - \mu_{\theta_1}(\mathbf{x}_t^{\theta_1}, t, y_1)}{\|\mathbf{x}'_{t-1} - \mu_{\theta_1}(\mathbf{x}_t^{\theta_1}, t, y_1)\|}, \tag{10}$$

*then, $\mathbf{x}_{t-1}^{\theta_1}$ is corrected onto $D_{t-1,y_1}^{\theta_1}$. Here, $\mu_{\theta_1}(\mathbf{x}_t^{\theta_1}, t, y_1)$ is defined by equation 6, $\eta_{t-1}^{\theta_1}$ is a small optimization step size.*

Proof is provided in Appendix A, and the geometry of manifold optimization is illustrated in Figure 3. Proposition 1 demonstrates that data can be corrected in a straightforward way to reside on $D_{t-1,y_1}^{\theta_1}$, thereby improving sampling quality. Furthermore, since the new variable contains information from $p_{\theta_2}$ and under Assumption 1, we can infer that it indeed resides on $D_{t-1,y_1}^{\theta_1} \bigcap D_{t-1,y_2}^{\theta_2}$, thus fulfilling the expectation of manifold optimization. We also introduce an adjustable aggregation step $s$, allowing flexible control over whether to incorporate information from other models to enhance the learned representations. Combining equations 9 and 10, the Aggregation of Two Diffusion Model algorithm is presented in Algorithm 1.

---

**Algorithm 1** Aggregation of Two Diffusion Models

---

**Input:** $t$-compatible model set $M = \{p_{\theta_1}, p_{\theta_2}\}$ under condition $Y = \{y_1, y_2\}$, aggregation step $s$, weighting factor $w$, optimization step $\eta_t^{\theta_1}$ and $\eta_t^{\theta_2}$
$\mathbf{x}_T^{\theta_1} = \mathbf{x}_T^{\theta_2} \sim N(\mathbf{0}, \boldsymbol{I})$
**for** $t$ in $[T:1]$ **do**
    $\mathbf{x}_{t-1}^{\theta_1} \sim p_{\theta_1}(\mathbf{x}_{t-1}^{\theta_1}|\mathbf{x}_t^{\theta_1}, y_1)$
    **if** $t > T - s$ **then**
        $\mathbf{x}_{t-1}^{\theta_2} \sim p_{\theta_2}(\mathbf{x}_{t-1}^{\theta_2}|\mathbf{x}_t^{\theta_2}, y_2)$
        $\mathbf{x}'_{t-1} = w\mathbf{x}_{t-1}^{\theta_1} + \sqrt{1-w^2}\mathbf{x}_{t-1}^{\theta_2}$
        $\mathbf{x}_{t-1}^{\theta_1} = \mathbf{x}'_{t-1} - \eta_t^{\theta_1} \frac{\mathbf{x}'_{t-1} - \mu_{\theta_1}(\mathbf{x}_t^{\theta_1}, t, y_1)}{\|\mathbf{x}'_{t-1} - \mu_{\theta_1}(\mathbf{x}_t^{\theta_1}, t, y_1)\|}$
        $\mathbf{x}_{t-1}^{\theta_2} = \mathbf{x}'_{t-1} - \eta_t^{\theta_2} \frac{\mathbf{x}'_{t-1} - \mu_{\theta_2}(\mathbf{x}_t^{\theta_2}, t, y_2)}{\|\mathbf{x}'_{t-1} - \mu_{\theta_2}(\mathbf{x}_t^{\theta_2}, t, y_2)\|}$
    **end if**
**end for**
**Output:** $x_0^{\theta_1}$

---

The algorithm comprises two key components: spherical aggregation and manifold optimization. Spherical aggregation aggregates the conditional control information from different models and ensures that the new intermediate variables remain stable near the manifold, while manifold optimization ensures more precise retention on the corresponding data manifold, enhancing sample quality. Note that each step of spherical aggregation also necessitates manifold optimization for $p_{\theta_2}$ to preserve the relevant conditional information within it, facilitating the subsequent aggregation step. This algorithm can be readily extended to multiple models, leading to the final Aggregation of Multi Diffusion Models (AMDM) algorithm, as shown in Algorithm 2.

The AMDM algorithm iteratively performs spherical aggregation and manifold optimization for each model during the first $s$ steps, followed by direct sampling from $p_{\theta_1}$. Moreover, since $\mu_{\theta_i}(\mathbf{x}_t^{\theta_i}, t, y_i)$ can reuse $\epsilon_{\theta_i}(\mathbf{x}_t^{\theta_i}, t, y_i)$ from the previous sampling step, the manifold optimization only introduces a single mathematical operation with no additional computational overhead, avoiding extra inference time.

## 4 EXPERIMENTS

In this section, we aggregate several representative models based on Stable Diffusion to evaluate the effectiveness of the algorithm. All experiments were conducted using a single RTX 3090 GPU.

**InteractDiffusion and MIGC.** InteractDiffusion (Hoe et al., 2024) is a T2I model that combines a pretrained Stable Diffusion (SD) model with a locally controlled interaction mechanism, enabling

---

**Algorithm 2** Aggregation of Multi Diffusion Models (AMDM)

---

**Input:** $t$-compatible model set $M = \{p_{\theta_1}, p_{\theta_2}, ..., p_{\theta_N}\}$ under condition $Y = \{y_1, y_2, ..., y_N\}$, aggregation step $s$, weighting factor $w_1, w_2, ..., w_{N-1}$ and optimization step $\eta_t^{\theta_1}, \eta_t^{\theta_2}, ..., \eta_t^{\theta_N}$

$\quad \mathbf{x}_T^{\theta_1} = \mathbf{x}_T^{\theta_2} = ... = \mathbf{x}_T^{\theta_N} \sim N(\mathbf{0}, \boldsymbol{I})$

$\quad$ **for** $t$ in $[T:1]$ **do**

$\quad\quad \mathbf{x}_{t-1}^{\theta_1} \sim p_{\theta_1}(\mathbf{x}_{t-1}^{\theta_1}|\mathbf{x}_t^{\theta_1}, y_1)$

$\quad\quad$ **if** $t > T - s$ **then**

$\quad\quad\quad \mathbf{x}_{t-1}^{\theta_i} \sim p_{\theta_i}(\mathbf{x}_{t-1}^{\theta_i}|\mathbf{x}_t^{\theta_i}, y_i)$ for $i$ in $[2:N]$

$\quad\quad\quad \mathbf{x}_{t-1}' = w_1 w_2 ... w_{N-1} \mathbf{x}_{t-1}^{\theta_1} + \sqrt{1-w_1^2} w_2 ... w_{N-1} \mathbf{x}_{t-1}^{\theta_2} + ... + \sqrt{1-w_1^2} ... \sqrt{1-w_{N-1}^2} \mathbf{x}_{t-1}^N$

$\quad\quad\quad \mathbf{x}_{t-1}^{\theta_i} = \mathbf{x}_{t-1}' - \eta_t^{\theta_i} \frac{\mathbf{x}_{t-1}' - \mu_\theta(\mathbf{x}_t^{\theta_i}, t, y_i)}{\|\mathbf{x}_{t-1}' - \mu_\theta(\mathbf{x}_t^{\theta_i}, t, y_i)\|}$ for $i$ in $[1:N]$

$\quad\quad$ **end if**

$\quad$ **end for**

**Output:** $x_0^{\theta_1}$

---

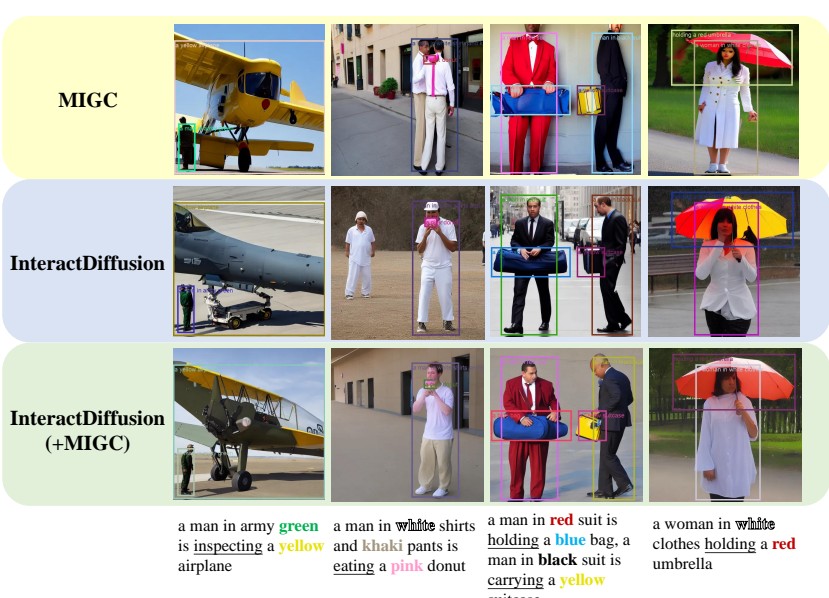

| | | | | |
|---|---|---|---|---|
| | a man in army **green** is inspecting a yellow airplane | a man in white shirts and khaki pants is eating a pink donut | a man in **red** suit is holding a **blue** bag, a man in **black** suit is carrying a yellow suitcase | a woman in white clothes holding a **red** umbrella |

Figure 4: Visual results of aggregating MIGC into InteractDiffusion applying the AMDM algorithm.

fine-grained control over the generated images and demonstrating effective interactivity. Similarly, MIGC (Zhou et al., 2024) is a T2I model that employs a divide-and-conquer strategy, achieving excellent performance in both attribute representation and isolation of generated instances.

InteractDiffusion primarily focuses on controlling subject-object interactions. However, due to the lack of explicit constraints on object attributes within the model architecture and dataset design, it exhibits suboptimal performance in attribute control. To address this, we attempt to aggregate the intermediate variables of MIGC $p_{\theta_2}$ into InteractDiffusion $p_{\theta_1}$ applying the AMDM algorithm, thereby introducing attribute control information into InteractDiffusion which we denoted as InteractDiffusion(+MIGC). The total sampling steps $T$ are set to 50, the aggregation step $s$ is set to 3, the weighting factor $w$ is set to 0.5 and the optimization steps $\eta_t^{\theta_1}$ and $\eta_t^{\theta_2}$ are simply set to 45 and 55, respectively. We use InteractDiffusion v1.0, and MIGC is modified to use the DDIM sampling method to align with the same diffusion process. Experimental results are shown in Figure 4. It is evident that aggregating the MIGC model into InteractDiffusion using our proposed AMDM algorithm significantly enhances its learned representations, leading to a notable improvement in instance attribute control, and confirming the algorithm's effectiveness.

| Method | Instance Success Rate (%) ↑ | | | | | | mIoU Score ↑ | | | | | | Time (s) |
|---|---|---|---|---|---|---|---|---|---|---|---|---|---|
| Level | $L_2$ | $L_3$ | $L_4$ | $L_5$ | $L_6$ | Avg | $L_2$ | $L_3$ | $L_4$ | $L_5$ | $L_6$ | Avg | |
| InteractiDiffusion | 31.87 | 26.66 | 23.90 | 23.37 | 23.85 | 24.96 | 27.99 | 24.13 | 21.45 | 21.51 | 21.16 | 22.44 | 18.76 |
| InteractiDiffusion(+MIGC) | **57.18** | **52.91** | **52.65** | **49.62** | **47.39** | **50.81** | **50.73** | **45.54** | **44.66** | **43.54** | **43.23** | **44.69** | 19.45 |

Table 1: Quantitative results on the COCO-MIG benchmark across different models. It demonstrates that aggregating MIGC into InteractiDiffusion leads to substantial improvements in all evaluated metrics.

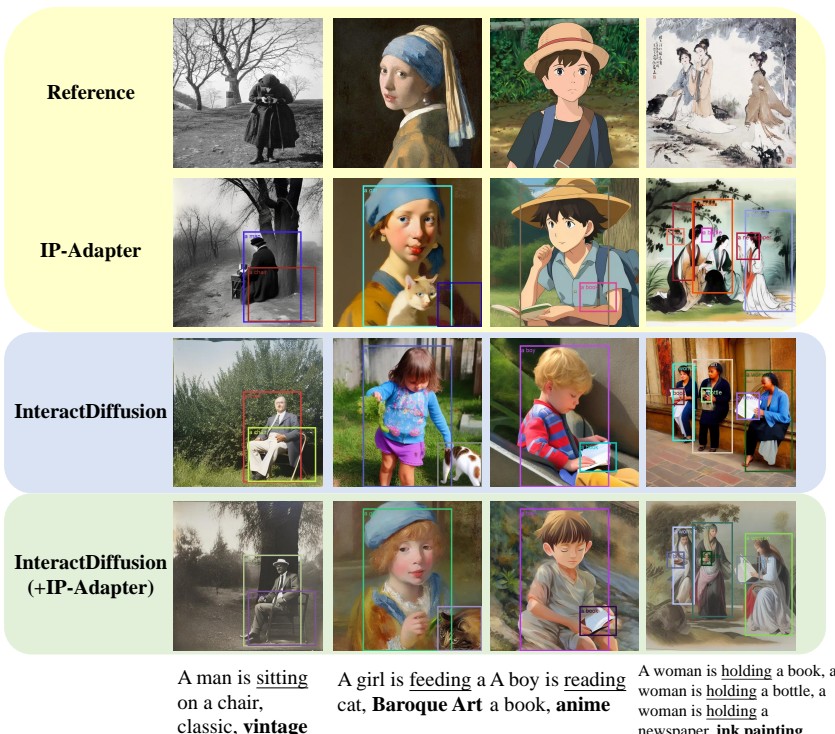

A man is sitting on a chair, classic, **vintage**

A girl is feeding a cat, **Baroque Art**

A boy is reading a book, **anime**

A woman is holding a book, a woman is holding a bottle, a woman is holding a newspaper, **ink painting**

Figure 5: Visual results of aggregating IP-Adapter into InteractDiffusion applying AMDM algorithm.

Currently, it is challenging to establish a comprehensive set of metrics and corresponding test sets to evaluate both attribute and interactive control performance of models. Therefore, we primarily focus on the improvement of aggregated control information metrics. For InteractDiffusion(+MIGC), we utilize the COCO-MIG Benchmark to assess the enhancement in attribute metrics. The COCO-MIG Benchmark employs the layout of COCO-position and assigns a specific color attribute to each instance, requiring that each generated instance not only satisfies positional requirements but also adheres to color attributes. The main process involves sampling layouts from COCO, filtering out smaller instances, and categorizing the layouts into five levels (L2-L6) based on the number of instances. Subsequently, a color is assigned to each instance within the sampled layout, selected from eight possible colors, and a global prompt is constructed, resulting in a test file with 800 entries. The COCO-MIG metrics primarily include Instance Success Rate and mIoU Score. The Instance Success Rate measures the probability of each instance being generated correctly, while mIoU Score calculates the average of the maximum IoU for all instances; if the color attribute is incorrect, the IoU value is set to 0.

Since the MIG-Benchmark does not contain interactive information, we set the "action" input in the InteractDiffusion model to "and". The final test results are shown in Table 1. It can be observed that the metrics for attribute control in InteractDiffusion(+MIGC) have significantly improved, further demonstrating the effectiveness of the algorithm.

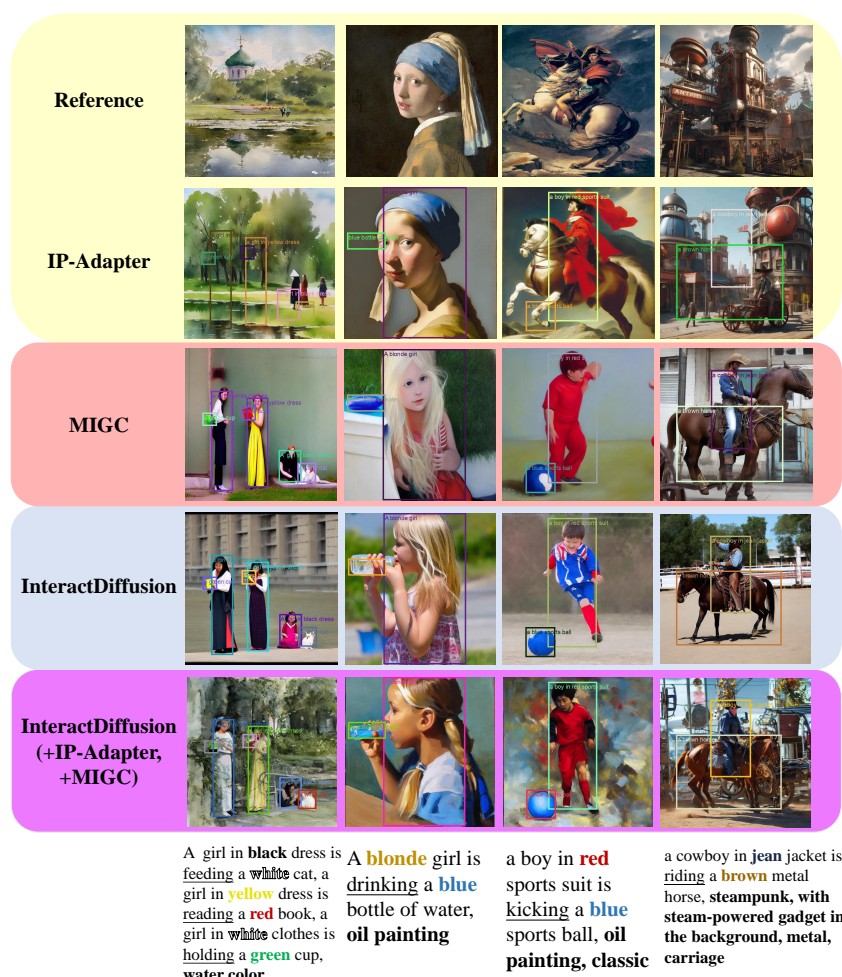

Figure 6: Visual results of aggregating MIGC and IP-Adapter into InteractDiffusion applying the AMDM algorithm.

**InteractDiffusion and IP-Adapter.** IP-Adapter (Ye et al., 2023) is a lightweight I2I model that employs a decoupled cross-attention mechanism to separately process text and image features, enabling multimodal image generation. Due to its superior performance in preserving the style of the reference image, we propose integrating the style information from IP-Adapter $p_{\theta_3}$ into InteractDiffusion $p_{\theta_1}$, denoted as InteractDiffusion(+IP-Adapter). The total sampling steps $T$ are set to 10, the aggregation step $s$ is set to 3, the weighting factor $w$ is set to 0.45, ip scale is set to 0.8 and the optimization steps $\eta_t^{\theta_1}$ and $\eta_t^{\theta_3}$ are simply set to 45 and 55, respectively. We utilized IP-Adapter based on SD1.5 while keeping InteractDiffusion unchanged. The experimental results are shown in Figure 5. It can be observed that IP-Adapter enhances the learned representations of InteractDiffusion, fully activating its style features, which further validates the effectiveness of the algorithm.

**InteractDiffusion, MIGC and IP-Adapter.** Furthermore, we attempt to aggregate the attribute features from MIGC $p_{\theta_2}$ and the style features from IP-Adapter $p_{\theta_3}$ into InteractDiffusion $p_{\theta_1}$ to evaluate the effectiveness of the AMDM algorithm. The pretrained models for the three architectures remain consistent with those mentioned above. The total sampling step $T$ set to 10, an aggregation step $s$ set to 3, and weight factors $w_1$ and $w_2$ set to 0.47 and 0.5. The optimization steps $\eta_t^{\theta_1}$, $\eta_t^{\theta_2}$, and $\eta_t^{\theta_3}$ are also simply set to 45, 40, and 35, respectively. The experimental results are presented in Figure 6.

These experiments provide robust evidence for the effectiveness of the AMDM algorithm. Furthermore, it can be inferred from the aggregation steps that the diffusion models initially focus on features such as position, attributes, and style, while later stages emphasize generation quality and consistency.

## 5 RELATED WORK

Classifier-free guidance in conditional diffusion models is widely applied in the field of image controlled generation. In addition to the classic directly text-driven models (Nichol et al., 2022; Ramesh et al., 2022; Rombach et al., 2022; Li et al., 2024b; Podell et al., 2024), an increasing number of studies are exploring more advanced and finer-grained conditional control techniques.

One of the most classical approaches in controllable generation is personalization controlled generation, which aims to capture and utilize complex concepts that are difficult to articulate through text. This method employs features such as style characteristics and attributes of subjects and objects from references as conditions for generation. Notable applications include style generation (Sohn et al., 2023; Liu et al., 2023; Hertz et al., 2024; Chen et al., 2024a), subject-driven generation (Ruiz et al., 2023; Li et al., 2024a; Shi et al., 2024; Jiang et al., 2024; Ye et al., 2023), person-driven generation (Xiao et al., 2024; Giambi & Lisanti, 2023; Valevski et al., 2023; Achlioptas et al., 2023; Peng et al., 2024), and interactive generation (Huang et al., 2023b; Guo et al., 2024; Wu et al., 2023b; Hoe et al., 2024). Additionally, spatial-controlled generation (Li et al., 2023; Cheng et al., 2023; Kim et al., 2023; Nie et al., 2024; Zhou et al., 2024) represents another significant research focus, primarily leveraging bounding boxes or various regions as additional input conditions to achieve specific spatial control objectives.

In recent years, several studies have attempted to achieve fine-grained control (Huang et al., 2023a; Han et al., 2023; Smith et al., 2023; Gu et al., 2024; Kumari et al., 2023) by designing various model architectures to handle inputs from various modalities, which require extensive training. These approaches inevitably necessitate a substantial amount of comprehensive multi-condition fine-grained data and the development of complex model architectures.

## 6 CONCLUSION

This paper proposes a novel AMDM algorithm, which consists of two main components: spherical aggregation and manifold optimization. Experimental results demonstrate the effectiveness of the AMDM algorithm, revealing that diffusion models initially prioritize image feature generation, shifting their focus to image quality and consistency in later stages. The algorithm provides a new perspective for addressing fine-grained conditional control generation challenge: We can leverage existing conditional diffusion models that control particular aspects, or develop and train new ones, and then apply the AMDM algorithm to achieve fine-grained control. This eliminates the need for constructing complex datasets, designing intricate model architectures, and incurring high training costs.

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

## A  PROOF OF PROPOSITION 1

**Proposition 1.** *For the diffusion model $p_{\theta_1}$ and any new intermediate variable $\mathbf{x}'_{t-1}$, let:*

$$\mathbf{x}^{\theta_1}_{t-1} = \mathbf{x}'_{t-1} - \eta^{\theta_1}_{t-1} \frac{\mathbf{x}'_{t-1} - \mu_{\theta_1}(\mathbf{x}^{\theta_1}_t, t, y_1)}{\|\mathbf{x}'_{t-1} - \mu_{\theta_1}(\mathbf{x}^{\theta_1}_t, t, y_1)\|}, \tag{10}$$

*then, $\mathbf{x}^{\theta_1}_{t-1}$ is corrected onto $D^{\theta_1}_{t-1,y_1}$. Here, $\mu_{\theta_1}(\mathbf{x}^{\theta_1}_t, t, y_1)$ is defined by equation 6, $\eta^{\theta_1}_{t-1}$ is a small optimization step size.*

*Proof*: Our objective is to correct the new variable $\mathbf{x}'_{t-1}$, which deviates from $D^{\theta_1}_{t-1,y_1}$, to ensure proper generation in subsequent sampling. Considering the step from $t$ to $t-1$, since $p_{\theta_1}(\mathbf{x}^{\theta_1}_{t-1} \mid \mathbf{x}^{\theta_1}_t, y_1)$ follows a normal distribution, the sampling process typically results in high-confidence clustering near the peak, causing the data to consistently reside in this region. Thus, we can move $\mathbf{x}'_{t-1}$ along the gradient of the probability density function $p_{\theta_1}(\mathbf{x}'_{t-1} \mid \mathbf{x}^{\theta_1}_t, y_1)$ by performing gradient ascent, thereby adjusting the value toward the peak region and aligning the data with the manifold $\mathcal{M} = \{\mathbf{x}^{\theta_1}_{t-1} \sim p_{\theta_1}(\mathbf{x}^{\theta_1}_{t-1} \mid \mathbf{x}^{\theta_1}_t, y_1)\}$. Since $\mathcal{M}$ is a submanifold of $D^{\theta_1}_{t-1,y_1}$, this optimization approach effectively pulls $\mathbf{x}'_{t-1}$ back onto $D^{\theta_1}_{t-1,y_1}$. Therefore, our optimization objective is:

$$\underset{\mathbf{x}^{\theta_1}_{t-1} \in \overline{U}(\mathbf{x}'_{t-1}, \delta)}{\arg\max} \quad [\nabla_{\mathbf{x}'_{t-1}} p_{\theta_1}(\mathbf{x}'_{t-1} \mid \mathbf{x}^{\theta_1}_t, y_1)]^T (\mathbf{x}^{\theta_1}_{t-1} - \mathbf{x}'_{t-1}), \tag{11}$$

where $\overline{U}(\mathbf{x}'_{t-1}, \delta) = \{\mathbf{x} \mid \|\mathbf{x} - \mathbf{x}'_{t-1}\| \leq \delta\}$ and $\delta$ is a small adjusting step, then:

$$\begin{aligned}
\mathbf{x}^{\theta_1}_{t-1} &= \mathbf{x}'_{t-1} + \delta \nabla_{\mathbf{x}'_{t-1}} p_{\theta_1}(\mathbf{x}'_{t-1} | \mathbf{x}^{\theta_1}_{t-1}) \\
&= \mathbf{x}'_{t-1} + \eta^{\theta_1}_{t-1} \frac{\nabla_{\mathbf{x}'_{t-1}} p_{\theta_1}(\mathbf{x}'_{t-1} | \mathbf{x}^{\theta_1}_t, y_1)}{\|\nabla_{\mathbf{x}'_{t-1}} p_{\theta_1}(\mathbf{x}'_{t-1} | \mathbf{x}^{\theta_1}_t, y_1)\|},
\end{aligned} \tag{12}$$

where $\eta^{\theta_1}_{t-1}$ is the step size for the unit gradient ascent. Since $p_{\theta_1}(\mathbf{x}'_{t-1} | \mathbf{x}^{\theta_1}_t, y_1)$ is a normal distribution, its gradient can be computed as follows:

$$\begin{aligned}
\nabla_{\mathbf{x}'_{t-1}} p_{\theta_1}(\mathbf{x}'_{t-1} | \mathbf{x}^{\theta_1}_t, y_1) &= \frac{1}{(2\pi\sigma^2_t)^{n/2}} \nabla_{\mathbf{x}'_{t-1}} \left( e^{-\frac{\|\mathbf{x}'_{t-1} - \mu_\theta(\mathbf{x}^{\theta_1}_t, t, y_1)\|^2}{2\sigma^2_t}} \right) \\
&= -\frac{1}{(2\pi\sigma^2_t)^{n/2}} e^{-\frac{\|\mathbf{x}'_{t-1} - \mu_\theta(\mathbf{x}^{\theta_1}_t, t, y_1)\|^2}{2\sigma^2_t}} \left( \frac{\mathbf{x}'_{t-1} - \mu_\theta(\mathbf{x}^{\theta_1}_t, t, y_1)}{\sigma^2_t} \right).
\end{aligned} \tag{13}$$

Simplifying the coefficient term, the final result is:

$$\mathbf{x}^{\theta_1}_{t-1} = \mathbf{x}'_{t-1} - \eta^{\theta_1}_{t-1} \frac{\mathbf{x}'_{t-1} - \mu_\theta(\mathbf{x}^{\theta_1}_t, t, y_1)}{\|\mathbf{x}'_{t-1} - \mu_\theta(\mathbf{x}^{\theta_1}_t, t, y_1)\|}, \tag{14}$$

which concludes the proof.

## B  ADDITIONAL VISUAL RESULTS

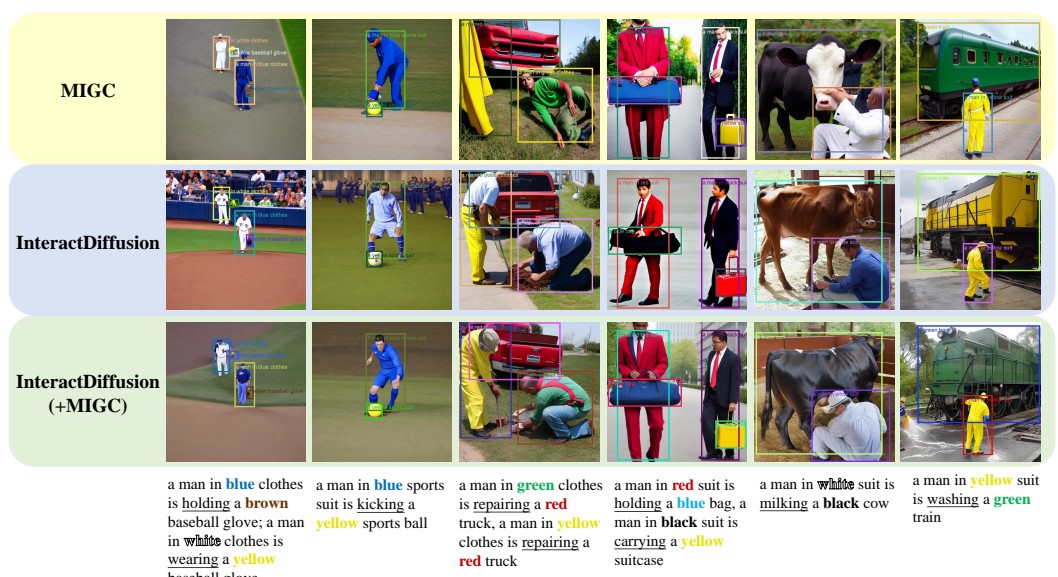

Figure 7: Additional Visual results of aggregating MIGC into InteractDiffusion applying the AMDM algorithm.

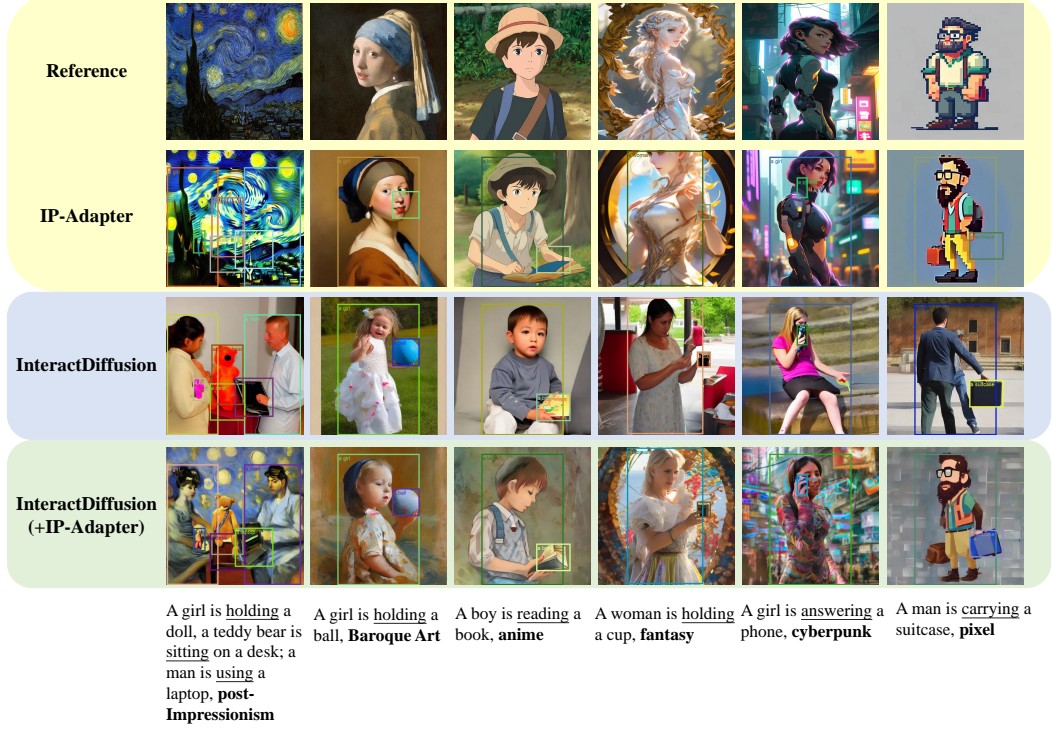

Figure 8: Additional Visual results of aggregating IP-Adapter into InteractDiffusion applying AMDM algorithm.

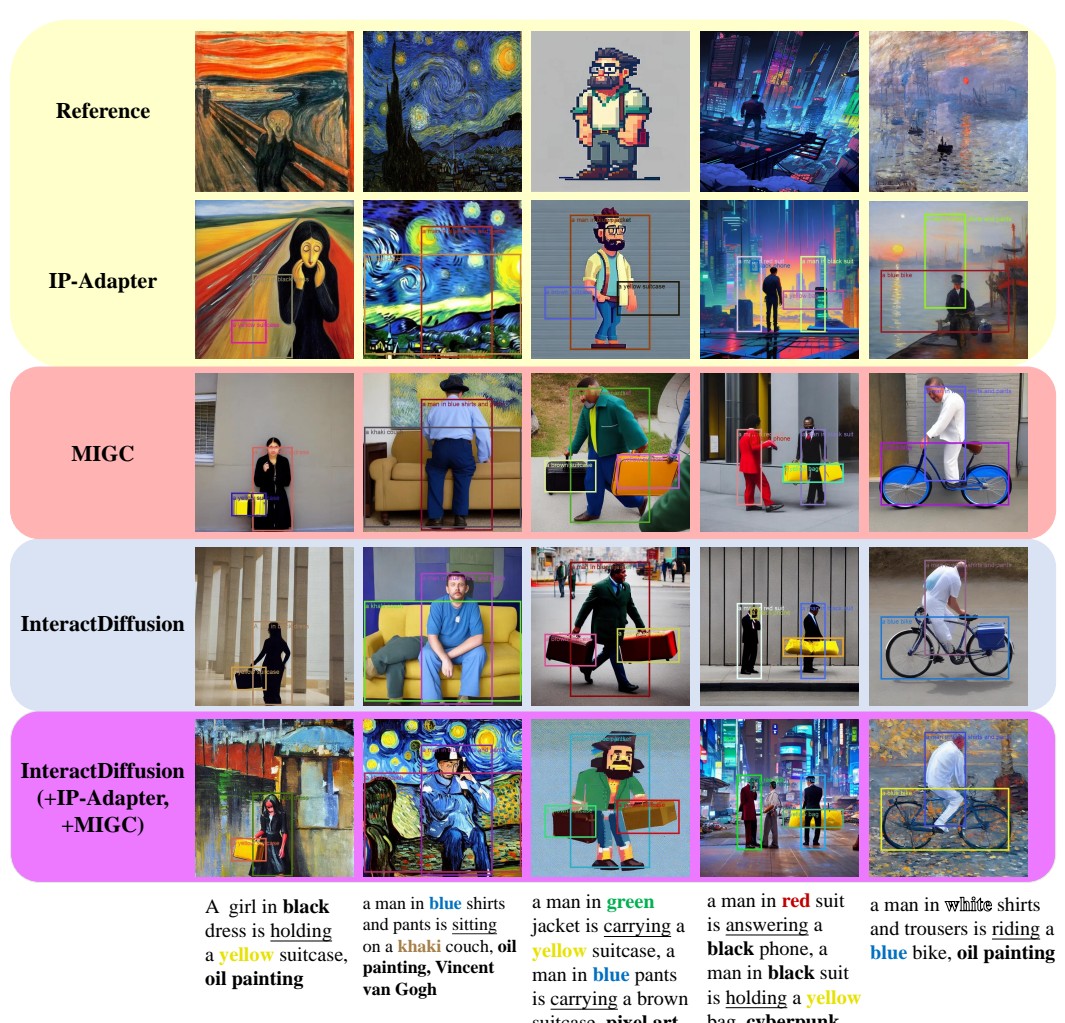

Figure 9: Additional Visual results of aggregating MIGC and IP-Adapter into InteractDiffusion applying the AMDM algorithm.

