# OpenReview forum: "Aggregation of Multi Diffusion Models for Enhancing Learned Representations"
_ICLR.cc/2025/Conference — ICLR 2025 Conference Withdrawn Submission_

### Official Review · Reviewer_Bxfn · 2024-11-01

**Soundness:** 2
**Presentation:** 2
**Contribution:** 2
**Rating:** 3
**Confidence:** 4

**Summary:**

This paper proposed to boost Stable Diffusion based image generation by aggregating multiple diffusion models. The authors interpolate the updates of different denoising models and then shift the interpolation toward the high-density area of one of the models. The authors show that such aggregation can improve the generation quality of existing T2I models.

**Strengths:**

The paper is mostly well-written and easy to follow. The experimental results match the authors' claim that AMDM combines the advantages of different models.

**Weaknesses:**

In general, I feel that the paper lacks a bit of novelty. The idea reminds me of previous works on the compositionality of diffusion models which are somewhat ignored in the paper. Some particular design choices are also not well-justified. See details below.

1. The manifold argument is a bit far-fetched. Eq. 9 is not spherical linear interpolation in general. Manifold optimization typically refers to numerical optimization on manifolds but here it is merely a buzzword. The definitions in Sec. 3.2 add very few to the paper - they are barely useful for the derivation and neither do they provide much insight.
2. Combining the updates of multiple diffusion models is not new. This has been extensively explored first in EBMs [1] and later in DMs [2]. Maybe there is some merit in using "spherical aggregation" specifically for combining these updates, but its necessity is not evident in the paper. For example, what if the authors just use linear interpolation? How will this affect the generation? Such comparison is important in this context.
3. Typos here and there. Multiple mistakes from lines 226 to 228. I don't know what is going on with Figure 2. There are other typos and I suggest the authors polish the paper.

[1] Du, Yilun, Shuang Li, and Igor Mordatch. "Compositional visual generation with energy based models." Advances in Neural Information Processing Systems 33 (2020): 6637-6647.

[2] Liu, Nan, et al. "Compositional visual generation with composable diffusion models." European Conference on Computer Vision. Cham: Springer Nature Switzerland, 2022.

**Questions:**

The image generation quality of the baseline methods seems a bit low. What are the SOTA methods currently in 2024? Have the authors consider works such as [3-4]? Will ADMD still be useful when the baselines are stronger?

[3] Mou, Chong, et al. "T2i-adapter: Learning adapters to dig out more controllable ability for text-to-image diffusion models." Proceedings of the AAAI Conference on Artificial Intelligence. Vol. 38. No. 5. 2024.

[4] Feng, Weixi, et al. "Training-free structured diffusion guidance for compositional text-to-image synthesis." arXiv preprint arXiv:2212.05032 (2022).

---

### Official Review · Reviewer_HkM2 · 2024-11-01

**Soundness:** 2
**Presentation:** 1
**Contribution:** 1
**Rating:** 3
**Confidence:** 4

**Summary:**

The authors propose a method for aggregating diffusion models of the same kind. It consists of two steps: spherical aggregation and "manifold optimization." Empirically, the authors demonstrate its effectiveness on multiple models and tasks, showing its superiority over individual models.

**Strengths:**

* I think that it easy to understand the method proposed by authors.
* The figures are neat and easy to understand.

**Weaknesses:**

* The biggest weakness of this work is the **lack of contextualization relative to prior work**. The authors do not mention any methods designed for composing generative models. I can suggest a few papers to start from: [5, 6, 7, 8]. The authors need to discuss those and other available methods for compositional generation, specifically:
    * How is their method different from others?
    * What potential unsolved problems it tackles?
    * **Importantly**: How does it compare experimentally?
* **Lack of ablation.** The authors propose some design choices like the choice of hyperparameters ($w, \eta_1, \eta_2, s$) and do not discuss how this choice was made and what impact does it have. I would like the authors to discuss the following:
    * How is the performance affected for different values of the above mentioned hyperparameters? How were they chosen?
    * **Importantly:** The authors propose spherical interpolation for combining information from different latent representations and argue that this is motivated by prior work [1]. How does this choice affect performance? How does it compare to an even simpler method like linear interpolation?
* **Limited contribution.** The authors summarize their contributions in Lines 088-101, which can be split into two: novel method for combining diffusion models of the same type and insights about diffusion models generative process. I will comment on each separately:
    * The authors do propose a method for combining diffusion models. There is an inherent limitation that this method only applies to diffusion models of the same kind (i.e. the same encoder, the same SDE, same noise schedule etc.). Furthermore we do not know if it is novel nor how it compares with other methods (see 1st weakness).
    * The authors claim that they make two observations about diffusion models in general:
        * First: "Diffusion models with a shared theoretical foundation possess the same mathematical essence, even if they differ in architecture, allowing operations on their intermediate variables". This is a trivial observation. This is because they approximate the same SDE. It has even been shown (Figure 7 in [2]) that latent representations of images coincide even if different model architectures are used to parametrize the score function. Furthermore, it is known (Figure 2 in [3]) that if you train two score-based diffusion models on two disjoint subsets of training data, they will generate the same image if conditioned on the same latent code, showing that it is the distribution of the data and the specification of the forward process that determine the generative distribution rather than specific architectures or data subsets.
        * Second: "diffusion models initially focus on the generation of features such as position, attributes, and style, while later stages emphasize generation quality and consistency.". First of all I do not believe that the authors demonstrate this apart from one sentence mentioning it in lines 487-489: "it can be inferred from the aggregation steps that the diffusion models initially focus on features such as position, attributes, and style, while later stages emphasize generation quality and consistency.". Second of all this is also a known observation discussed e.g. in [4].
* **Lack of mathematical rigor.** For example:
    * Definition 1 -  This definition is not a precise mathematical definition. I would like the authors to clarify what they mean by $D$. Is this supposed to be the support of the distribution $p(x_t|y)$? Or the typical region of this distribution [9]? According to the explanation below "all possible data generated by a diffusion process". This is not a useful definition, because for any $t > 0$ this is the whole of $\mathbb{R}^n$. This is because we are convolving the data distribution with another distribution whose support is $\mathbb{R}^n$. I presume that the authors rather meant the typical set, but this needs to be clarified.
    * Line 195 The authors refer to [1] to claim that $D$ will reside on an $(n-1)$ manifold. I am confused by how this is relevant. [1] assumes that data lies in an $l$-dimensional linear subspace. Are the authors making the same assumption? Also, [1] proves that the "noisy" manifold is "concentrated" on some manifold, where "concentration" is very precisely defined. Do the authors share the same definition when talking about the definition of $D$? This paragraph is not rigorous and more confusing rather than helpful.
    * Line 230: What is the "latent space encoder"? This has not been properly defined. I assume the authors mean different latent diffusion model, but sharing the VAE encoder? I.e. the mapping from data to $z_0$? Are $x_t$ latent? I do not think that latent diffusion was actually defined in the paper mathematically.
    * Line 262: Authors write "A Gaussian sample is likely to be drawn near the peak" Since this is not a precise statement, I assume that the authors mean that all samples are concentrated near the mean and this is a false statement. See [9] for an explanation.
    * Misuse of the term "manifold". For example: The "manifold optimization" step proposed by the authors is essentially shifting a sample closer to the mean of the Gaussian. What is the manifold there? I think that this is a simple procedure that is described in unnecessarily complicated terms.

---

References

[1] Chung et al. "Improving Diffusion Models for Inverse Problems using Manifold Constraints" (NeurIPS 2022)

[2] Song et al. "Score-Based Generative Modeling through Stochastic Differential Equations" (ICLR 2021)

[3] Kadkhodaie et al. "Generalization in diffusion models arises from geometry-adaptive harmonic representations" (ICLR 2024)

[4] Deja et al. "On Analyzing Generative and Denoising Capabilities of Diffusion-based Deep Generative Models" (NeurIPS 2022)

[5] Du et al. "Compositional visual generation with energy based models" (NeurIPS 2020)

[6] Du et al. "Reduce, reuse, recycle: Compositional generation with energy-based diffusion models and mcm" (ICML 2023)

[7] Garipov et al. "COMPOSITIONAL SCULPTING OF ITERATIVE GENERATIVE PROCESSES" (NeurIPS 2023)

[8] Du et al. "Compositional Generative Modeling: A Single Model is Not All You Need" (ICML 2024)

[9] Nalisnick et al. "Detecting Out-of-Distribution Inputs to Deep Generative Models Using Typicality" (arXiv)

**Questions:**

* Line 149 "analyzes" -> "analyze"
* Line 256: Do the deviations happen in practice?
* Lines 264-268: The authors mention that they propose to shift the intermediate latent point by performing gradient ascent wrt $p_{\theta_1}(x_{t-1}|x_t)$. First of all, since this is a Gaussian we know it in closed-form so why use gradient ascent and not just target the specific value of the density? Secondly, I think that the authors meant the log-density and not density (correct me please if I'm wrong)? Thirdly: When I look at the proof I do not see the authors actually using gradient ascent. In the proof I do not understand where Equation (11) came from. How is it derived? Why is this the objective?
* Line 280: What does it mean that the "new variable contains information from $p_{\theta_2}$"? Can the authors make this statement more precise?

---

### Official Review · Reviewer_xgE4 · 2024-11-03

**Soundness:** 3
**Presentation:** 3
**Contribution:** 3
**Rating:** 5
**Confidence:** 5

**Summary:**

The paper studies the problem of aggregating multiple diffusion models for more fine-grained control of the generated results. The paper proposes two steps: 1) use spherical aggregation to mix the latent diffusion representations and 2) use manifold optimization to bring the aggregated result to samples with high probability. The paper shows that the proposed method can achieve better control by mixing two diffusion models with different specialties.

**Strengths:**

1. The paper proposes a method for effectively mixing the generation process of two or more diffusion models without extra training.

2. The paper shows that the generated results can be better controlled by combining diffusion models with different specialties.

**Weaknesses:**

1. Some basic metrics of the aggregated performance should also be provided, e.g., DINO, FID, and CLIP score, and compared to the model outputs without aggregations.

2. No ablation study result is provided. How should the performance change if we use other aggregation methods (e.g., linear aggregation), if we do not use the manifold optimization, or if we use the manifold optimization with various hyperparameters?

3. Compared to the other approach, where training is applied to improve the fine-grained control of the diffusion model generation, the proposed approach requires incorporating multiple diffusion models during inference, increasing the inference costs. Including results from those methods as a reference is also beneficial. It may also result from some more advanced foundational models that are known to follow text better, such as SD3 and FLUX.

**Questions:**

1. Can you clarify the statement in Lines 280-282 about the manifold optimized point that should lie in the intersection?

2. It seems that one of the diffusion models has to be chosen as theta_1. How does the performance change if we change the choice of theta_1 in the experiments? Given an arbitrary set of diffusion models, how should the user choose theta_1?

---

### Official Review · Reviewer_pM84 · 2024-11-04

**Soundness:** 3
**Presentation:** 3
**Contribution:** 3
**Rating:** 6
**Confidence:** 4

**Summary:**

This paper introduced a new method to aggregate different diffusion models in order to increase the generated image quality and the adherence to prompt and instruction (such as localization). This new algorithms spherically interpolates intermediary samples of the diffusion process from different diffusion models up until a chosen timestep (at which point semantic information is often already set and will not change during the rest of the diffusion process). The authors present some theoretical elements about rectifications made after spherically interpolating images. Final images look of higher quality than original ones while also better following conditionning elements.

**Strengths:**

* To my knowledge the method and idea are novel. They provide a cheap method to improve diffusion model sampling without any retraining step.
* Visually, the method seems to be improving both prompt adherence and image quality.
* There is an appropriate amount of theory derivation to explain the different steps of the algorithm.

**Weaknesses:**

* The paper is very light in quantitative metrics about image quality. The only two metrics seem to be about prompt adherence but there is nothing about image quality.
* The paper lacks some comparison with existing methods (e.g diffusion model ensembling)
* Compared to model merging, this method stills incurr an additional computational overhead when sampling with 2 models (twice the amount of Neural Function Evaluation).

**Questions:**

1. Could you provide comparisons with score function ensembling?
2. How does the method scale to more than 3 diffusion models? Is there a limit at wich point intermediary samples are averaged too much and the backward process fails at producing a good quality image?
3. Does the added "robustness" of having several models allow for a lower amount of diffusion steps?
4. How important is the hyperparameter $s$? Does going all the way with $s=T$ improves the generated images? What is the minimal amout of "merging" compared to just following a single diffusion model?
5. Your method works on merging intermediary samples of the diffusion process. I guess the underlying architecture of the different diffusion models can be very different (apart from the shared latent encoder). Do the different diffusion models you use share an architecture? If yes, do you think model merging (where weights of different versions of the same model are interpolated together) would work?

---

### Official Review · Reviewer_zkvQ · 2024-11-04

**Soundness:** 3
**Presentation:** 3
**Contribution:** 3
**Rating:** 3
**Confidence:** 3

**Summary:**

This paper studies combining multiple diffusion models, using different features from each, to form one final model. The goal is to get more fine-grained control, overcoming limitations of existing guidance methods. The new algorithm Aggregation of Multi Diffusion Models (AMDM) consists of two key components: spherical aggregation and manifold optimization. The authors show that AMDM can improve fine-grained control and that they can use conditional diffusion models for specific aspects while aggregating the models. This avoids the need for custom datasets for each aspect the user may want to control.

**Strengths:**

The authors show the benefits of combining multiple diffusion models, as a way to get fine-grained control over multiple aspects. This is an interesting approach that could open up a lot of use cases for diffusion models, without needing to re-train or to train custom adapters for each style, attribute, etc.

The paper provides a theoretical justification of how their method works by analyzing the diffusion process. The authors are very clear about the technical details of their Spherical Aggregation and Manifold Optimization steps.

The authors provide both qualitative and quantitative evidence for the applicability and success of their method. For example, results on the COCO-MIG benchmark show improvements in several metrics.

**Weaknesses:**

* The authors focus mostly on COCO-style captions and images. Hence, the evaluation is mostly on short captions and photo-realistic images of everyday life. However, many use cases of guidance go beyond these domains. It is not clear if this method can help with generating designs, logos, text, other types of digital art, etc. The study would be more convincing if the authors could evaluate at least 1-2 other domains.

* The paper is light on baselines. For example, Table 1 only shows the authors' algorithms. How do the results compare to the several models mentioned in the related work, e.g.,  "several studies have attempted to achieve fine-grained control (Huang et al., 2023a;
Han et al., 2023; Smith et al., 2023; Gu et al., 2024; Kumari et al., 2023)....." I understand that these methods may require training and/or architecture changes. But it would be good to know how AMDM compares to other approaches. Concretely, Layoutdm, Deadiff and Animatediff should be included for comparison / combination with AMDM. Or explain why direct comparisons to these methods may or may not be appropriate given the different approaches (e.g., training requirements, architectural differences).

* Typo: In table 1, it says "InteractiDiffusion" instead of "InteractDiffusion"

**Questions:**

* The abstract says "Code is available at: https://github.com/Hammour-steak/AMDM " which seems to violate the anonymous policy.

* The paper seems to only present experiments on integrating into InteractDiffusion. Do the results hold for other models / combinations?

* The intro states a number of nice issues "Generating multiple objects with overlapping bounding boxes can lead to attribute leakage, where one object’s description inappropriately influences others, causing inconsistencies between objects and the background. Fine-grained interaction details may be illogical, and style integration may compromise object attributes." However, the paper does not seem to discuss these issues and whether AMDM can help solve them.

* I didn't quite follow where in the paper this contribution is discussed/justified: "Our algorithm and experiments reveal some unique properties of diffusion models: Diffusion models with a shared theoretical foundation possess the same mathematical essence,
even if they differ in architecture, allowing operations on their intermediate variables; Furthermore, diffusion models initially focus on the generation of features such as position, attributes, and style, while later stages emphasize generation quality and consistency." But I would be interested in understanding this better, as it seems like a deep contribution. Specifically, can you give detailed explanations or experimental evidence supporting these claims about the properties of diffusion models, and clarification on how these properties are demonstrated through the AMDM algorithm and experiments.

---

### Official Review · Reviewer_ZCg9 · 2024-11-08

**Soundness:** 3
**Presentation:** 3
**Contribution:** 3
**Rating:** 1
**Confidence:** 5

**Summary:**

This submission does not comply with the double-blind review policy since the provided code repo is non-anonymous.

**Strengths:**

This submission does not comply with the double-blind review policy since the provided code repo is non-anonymous.

**Weaknesses:**

This submission does not comply with the double-blind review policy since the provided code repo is non-anonymous.

**Questions:**

N/A

---

### Official Review · Reviewer_jabV · 2024-11-08

**Soundness:** 2
**Presentation:** 3
**Contribution:** 2
**Rating:** 5
**Confidence:** 3

**Summary:**

The paper introduces the Aggregation of Multi Diffusion Models (AMDM), an algorithm to enhance fine-grained control in image generation by combining features from multiple diffusion models. It leverages two main techniques: spherical aggregation, which merges intermediate features with minimal manifold deviation, and manifold optimization, which refines these variables to align with the intermediate data manifold.

**Strengths:**

- New insights where the diffusion model fails to capture certain aspects of features
- Good performance in the downstream benchmarking

**Weaknesses:**

- A computational complexity comparison is needed during the inference to evaluate the algorithm's associated tradeoff, such as inference time.
- Integrating multiple models might introduce unanticipated effects, especially when merging highly distinct models. Although manifold optimization aims to correct deviations, there’s little discussion on potential artifacts in complex or real-world scenarios.

**Questions:**

- Is AMDM applied solely during inference with pre-trained models, or do you simultaneously train the diffusion models alongside the aggregation process?
- Have you experimented with non-linear interpolation techniques, such as splines, rather than linear interpolation? Understanding how different interpolation methods might impact downstream performance would be insightful.
- Has AMDM been evaluated on tasks where achieving fine-grained control is especially challenging, such as aggregating models with contrasting styles (e.g., realism and abstract)?
- It would be valuable to visualize or analyze the learned weighting factors and the types of weighting factors learned during aggregation, as this could offer insights into the underlying structure of the intermediate variable space.

**Details Of Ethics Concerns:**

Potential breach of double-blind review:


The GitHub link in the paper https://github.com/Hammour-steak/AMDM is from a personal repository: https://github.com/Hammour-steak.

---

### Note · Authors · 2024-11-24

I have read and agree with the venue's withdrawal policy on behalf of myself and my co-authors.